acoustics/behaviour/biogeography

passive acoustic monitoring, cultural transmission, humpback whale, migration, vocal learning

**Author for correspondence:**
Victoria E. Warren
e-mail: vwar775@aucklanduni.ac.nz

# Migratory insights from singing humpback whales recorded around central New Zealand

Victoria E. Warren[1,2], Rochelle Constantine[1,3],
Michael Noad[4], Claire Garrigue[5,6] and Ellen C. Garland[7]

[1]Institute of Marine Science, Leigh Marine Laboratory, University of Auckland, 160 Goat Island Road, Leigh 0985, New Zealand
[2]National Institute of Water and Atmospheric Research, 301 Evans Bay Parade, Hataitai, Wellington 6021, New Zealand
[3]School of Biological Sciences, University of Auckland, 3A Symonds Street, Auckland 1010, New Zealand
[4]Cetacean Ecology and Acoustics Laboratories, School of Veterinary Science, The University of Queensland, Australia
[5]UMR Entropie (IRD, Université de La Réunion, Université de la Nouvelle-Calédonie, IFREMER, CNRS) BP A5, 98848 Nouméa, New Caledonia
[6]Opération Cétacés, 98802 Noumea, New Caledonia
[7]Sea Mammal Research Unit, Scottish Oceans Institute, School of Biology, University of St Andrews, Fife KY16 8LB, UK

VEW, 0000-0001-9040-4831; RC, 0000-0003-3260-539X;
MN, 0000-0002-2799-8320; CG, 0000-0002-8117-3370;
ECG, 0000-0002-8240-1267

The migration routes of wide-ranging species can be difficult to study, particularly at sea. In the western South Pacific, migratory routes of humpback whales between breeding and feeding areas are unclear. Male humpback whales sing a population-specific song, which can be used to match singers on migration to a breeding population. To investigate migratory routes and breeding area connections, passive acoustic recorders were deployed in the central New Zealand migratory corridor (2016); recorded humpback whale song was compared to song from the closest breeding populations of East Australia and New Caledonia (2015–2017). Singing northbound whales migrated past New Zealand from June to August via the east coast of the South Island and Cook Strait. Few song detections were made along the east coast of the North Island. New Zealand song matched New Caledonia song, suggesting a migratory destination, but connectivity to East Australia could not be ruled out. Two song types were present in New Zealand, illustrating the potential for easterly song transmission from East Australia to New Caledonia in this shared migratory corridor. This

study enhances our understanding of western South Pacific humpback whale breeding population connectivity, and provides novel insights into the dynamic transmission of song culture.

## 1. Introduction

Large-scale animal movements, known as migrations, are commonly exhibited when a spatial disparity exists in the location of resources [1]. Migratory routes between resources can be difficult to study for wide-ranging animals, particularly those in the marine environment that can be challenging to locate and track. Humpback whales (*Megaptera novaeangliae*) undertake some of the longest migratory journeys of any mammal [1,2], due to spatial disparity in preferred feeding and breeding grounds. In the South Pacific, humpback whales mate and calve during the austral winter at breeding grounds that shelter genetically distinct populations [3–5]. Some transience and interchange of individuals has been observed among breeding areas [6–11], but most animals return to the same location each year [7] due to maternally driven site fidelity [12], in spite of a lack of geographical barriers between breeding areas [4].

During the austral summer, humpback whales feed at high latitudes. Historically, South Pacific whales were thought to feed in Antarctic areas roughly south of their breeding grounds [13,14]. However, recent studies using satellite tags, photographic mark–recapture and genetic analyses have revealed that humpback whales spread across a vast range of circumpolar longitudes during the feeding season and mix with whales from other breeding areas [2,5,11,15–22]. This complexity in movement, in conjunction with limited resources to study their lengthy migration patterns, results in a paucity of understanding about the migratory routes of humpback whales that breed in the South Pacific.

New Zealand lies between the feeding and breeding grounds of western South Pacific humpback whales, and seasonally hosts migrating whales. Historically, humpback whales migrated northbound past mainland New Zealand between May and August, following multiple routes to their breeding areas [23]. An individual's choice of migration route could be influenced by the location of the feeding ground from which it is coming [14], the breeding ground to which it is going [14,24] or its demography (such as sex, age class or reproductive status) [21,25,26]. The seasonal presence of humpback whales around mainland New Zealand was historically exploited by land-based whaling operations [23]. Following the collapse of South Pacific humpback whale populations due to extensive hunting pressure, New Zealand's whaling stations closed in the 1960s [27]. Recovery of western South Pacific humpback whale populations has since occurred, to differing extents [3,28,29], and increasing numbers of humpbacks are now observed during migration past mainland New Zealand [30,31]. A range of research methods have revealed linkages between humpbacks migrating through New Zealand waters to breeding areas in East Australia and western Oceania (namely New Caledonia, Fiji and Tonga) [11,13,14,18,22,24,31–35].

During the southbound migration, between September and December, humpback whales are less commonly sighted around mainland New Zealand [23,24,30]. Recently, it was found that at the most northerly extent of New Zealand, the Kermadec Islands, large numbers of southbound humpback whales converge from multiple Oceanian breeding grounds before proceeding south and southeast to Antarctic feeding grounds, without passing mainland New Zealand [22,35–37]. Satellite tag data have also revealed that southbound East Australian humpback whales generally do not travel via New Zealand, except for a few that cross the Tasman Sea to the west coast of New Zealand's South Island before turning south to Antarctica [18,32].

In addition to satellite tracking, and photographic and genotype identification, humpback whales can be linked to their respective breeding grounds via acoustic data. Male humpback whales produce cyclic, stereotyped vocalizations with hierarchical structure, known as 'song' [38]. Song is composed of sound 'units', which are produced in a stereotyped sequence to make a 'phrase'; phrases are repeated multiple times to produce a 'theme', and multiple different themes are sung in a stereotyped order to create a 'song cycle'. Song commonly differs between breeding grounds each year during the breeding season, and, in general, all males on a breeding ground sing the same version of a song (termed 'song type') [39–41]. Robust, repeatable analyses are available for quantitative song matching [42]. The song produced by a male humpback whale can indicate which breeding population it is most likely associated with within a given year [37,39,43]. Therefore, acoustic monitoring provides a cost-effective, broad-scale, all-weather and long-term study methodology that can expand on results obtained using other methods.

Humpback whales are 'vocal production learners', meaning that they can modify their acoustic signals following exposure to other signals [44]. In the South Pacific, there is an eastward transmission of song between breeding grounds over consecutive years [39], which is assumed to be the result of cultural transmission. However, as with other humpback whale populations, breeding grounds in the South Pacific are acoustically isolated from each other due to the distances between them and it is not possible for whales to hear songs sung at other breeding grounds. Payne & Guinee [45] proposed that song transmission could occur when individuals moved between breeding grounds within a year, when individuals moved between breeding grounds between years, or on shared feeding grounds, and/or on shared, or partially shared, migration routes. It is not uncommon to record humpback song outside of breeding grounds [19,38,46,47], including during migration past mainland New Zealand [40,48,49]. Given that the migratory corridor through New Zealand waters has been connected to multiple breeding areas in the western South Pacific [11,13,14,22,24,31–35], it is feasible that song transmission could occur around mainland New Zealand.

Here, we examined humpback whale song detected in passive acoustic data collected in central New Zealand during 2016, and in conjunction with song recorded on western South Pacific breeding grounds in 2015–2017, we aimed to address four questions: (i) When is humpback whale song detected in central New Zealand waters? (ii) Which migration route(s) do singing humpback whales take around central New Zealand? (iii) What is the likely breeding ground destination of these whales, as suggested by song? (iv) Do the data help to explain when or where song may be culturally transmitted? The results of this study will expand upon recent land-based survey information about humpback whale occurrence around mainland New Zealand, and improve our broader understanding of complex humpback whale migrations and dynamic song transmission in the western South Pacific.

# 2. Material and methods

## 2.1. Song recordings

There are three principal migratory routes for humpback whales passing central New Zealand [23]. Some whales migrate north along the east coasts of the South and North Islands, others remain further offshore and to the west of both islands, and some pass through Cook Strait in central New Zealand, which, in a northbound direction, connects the east coast of the South Island to the west coast of the North Island (figure 1). In order to capture the different routes through central New Zealand, four autonomous multi-channel acoustic recorders (AMARs, JASCO Applied Sciences) were deployed around central New Zealand from 4 June to 21 December 2016 (figure 1, inset). Off the coasts of Kaikōura (42.31° S, 174.21° E) and Wairarapa (41.61° S, 175.90° E), AMARs were moored approximately 10 m from the seabed at depths exceeding 1 km. In the South Taranaki Bight (STB) (40.42° S, 174.50° E) and Cook Strait (41.09° S, 174.55° E), recorders were bottom-mounted in water depths shallower than 300 m. AMARs sampled with a duty cycle of 900 s: 630 s at 16 kHz (WAV format, 24 bit), 125 s at 250 kHz (WAV format, 16 bit) and 145 s of sleep.

New Zealand data were collected during 2016; thus song from the closest breeding populations was obtained for the year prior (2015), the same year (2016) and the year after (2017). East Australian song was recorded approximately 1.5 km off Peregian Beach, Queensland (26.48° S, 153.10° E) [39,50,51]. While this is a migratory corridor, around 600 km south of the East Australian breeding grounds around the Great Barrier Reef [52] (figure 1), there is extensive evidence that breeding activities also occur near Peregian Beach [25,51,53,54]. East Australian data were recorded using two Acousonde recorders (Acousonde 3-A with external battery housings, Greenridge Sciences) moored 4 m from the seabed in 25 m of water, with a sampling rate of 25.8 kHz, 16 bit, and 9 kHz low-pass filter. Recordings were made using lossless compression, and were later decompressed to WAV files. The two recorders featured alternating 12 h duty cycles, resulting in continuous recording. Recordings were made during both the northbound and southbound migrations (2015), southbound only (September–October; 2016) or northbound only (June–July; 2017).

Song was also available from the New Caledonia breeding ground for 2015, 2016 and 2017 (figure 1). Song was recorded using a moored recorder (SM2M + Whalesong recorder, Wildlife Acoustics, SMX-II microphone) deployed at approximately 60 m depth, between July and September in the southern lagoon of New Caledonia (22.5° S, 167.0° E). Data were recorded for 18 h of every 24 h, with a sampling rate of 22 kHz, and 3 Hz high-pass filter. As with the Acousonde recorders, recordings were made using a proprietary lossless compression format and later converted to 16 bit WAV files.

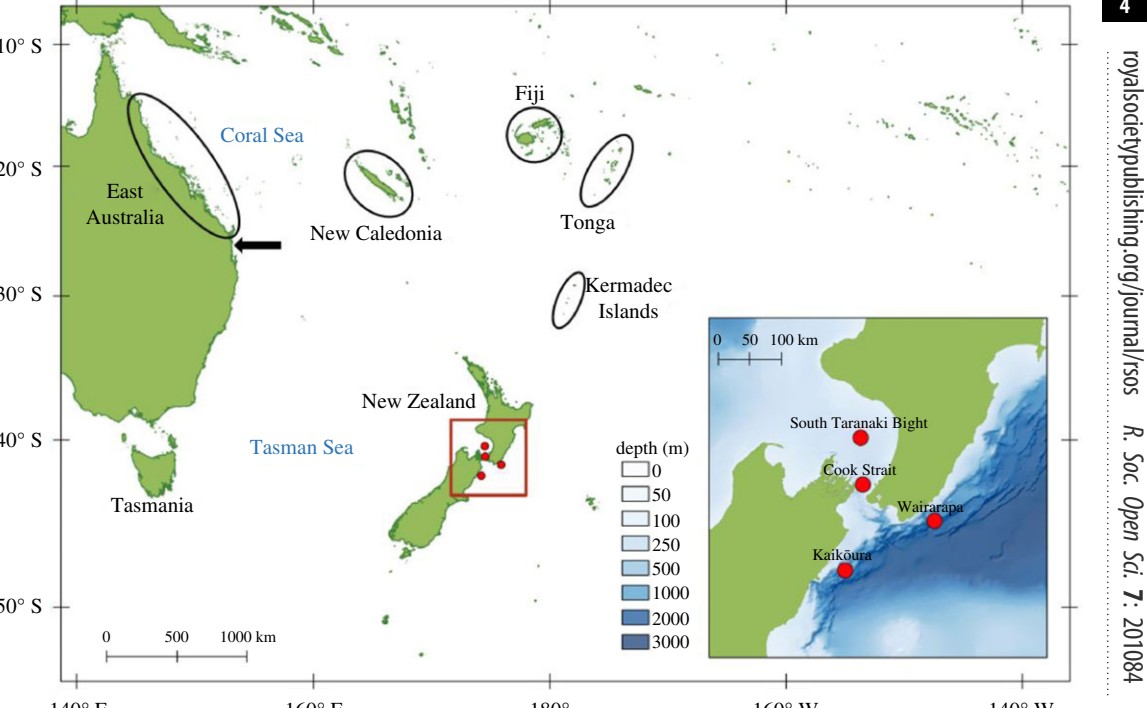

**Figure 1.** Map of the western South Pacific, indicating humpback whale breeding grounds (East Australia, New Caledonia, Fiji and Tonga) and the location of the Kermadec Islands. The black arrow indicates the location of Peregian Beach where the East Australian recordings were made. Inset: acoustic recording locations around central New Zealand, illustrated as red circles.

## 2.2. Detection of song in New Zealand data

In order to identify humpback whale song in the New Zealand data, the PAMGuard whistle and moan detector [55,56] was applied to all 16 kHz data from the four recording locations (full details provided in electronic supplementary material, S1). The detector outputs were used to direct in-depth examination of the data; data from time periods with detections were thoroughly manually examined to determine the presence or absence of humpback whale song. All 16 kHz files containing PAMGuard detections were opened and viewed as spectrograms in Raven Pro [57] with FFT length 1024, Hann window with 75% overlap. Files were categorized as 0 (no song), 1 (song evident, but with low signal-to-noise ratio (SNR)) or 2 (high SNR song, suitable for analysis). The SNR of song in category 2 was approximately 10–20 dB above background noise. Song presence was classified as any file containing song, regardless of quality, and was used to calculate the percentage of files per day containing song, per recording location. In order to identify the start and end of humpback song presence in the data, files recorded before and after those with PAMGuard detections were also checked. Humpback whale song, as a proxy for vocal animal presence, was deemed to have ceased when at least five full days of recordings contained no song at any recording location.

## 2.3. Song transcription

New Zealand data that contained high-quality song (i.e. category 2) were further analysed to transcribe the sequence of units. Each song unit was ascribed a descriptive name based on its subjective features following previous studies [42,58]. In the New Zealand data, all high-quality song from all recording locations was transcribed (table 1). In addition to song detected in the 16 kHz data, song recorded during the intervening 250 kHz duty cycle was also transcribed to maximize the quantity and duration of transcribed song. The 250 kHz files were opened and viewed in Raven Pro with FFT length 16 384, Hann window with 75% overlap. Song samples in the New Zealand data were 40–85 min in length, comprising several duty cycles of recording; song was analysed within the constraints imposed by the non-recording period of each duty cycle. Within a passive acoustic monitoring framework, it is not possible to know whether song was produced by multiple

**Table 1.** Transcribed New Zealand 2016 song data from two locations resulting in six recording events (individuals). The recording ID (X, Y or Z) identifies each analysed song session per location/year combination. Recording location STB = South Taranaki Bight. Duration = duration of transcribed song, inclusive of duty cycled non-recording periods (145 s per 900 s).

| recording location | year | recording ID | date | duration (min) | migration stage |
|---|---|---|---|---|---|
| STB | 2016 | X | 2 July | 55 | northbound |
| STB | 2016 | Y | 5 July | 55 | northbound |
| STB | 2016 | Z | 21 July | 60 | northbound |
| Cook Strait | 2016 | X | 10 July | 60 | northbound |
| Cook Strait | 2016 | Y | 22 July | 40 | northbound |
| Cook Strait | 2016 | Z | 9 Aug | 85 | northbound |

consecutive males, or by one animal that remained in a locale and sang repeatedly. Here, high-quality song was recorded with several days separation at each New Zealand recording location, thus, each song sample was treated as a different, individual whale.

Songs were also transcribed for New Caledonia and East Australia for each year, when it was possible to consistently identify one singing individual for at least 30 min (table 2). Songs were transcribed in Raven Pro (using the same parameter values as the New Zealand 16 kHz data) for a maximum of 60 min. Three song sessions were transcribed per location and per year. Owen *et al*. [37] previously identified two song types in the New Caledonia breeding ground in 2015, and both of these were transcribed for this study (henceforth labelled A and B), making a total of six transcribed songs for New Caledonia in 2015 (table 2). Where possible, within the constraints of high-quality song recordings, the transcribed East Australian and New Caledonian songs were recorded at a similar time of year as the recordings made in New Zealand.

## 2.4. Quantifying unit classification using random forest

A total of 71 unit types were qualitatively described during transcription of the New Zealand, New Caledonia and East Australia songs (electronic supplementary material, table S2.1). To ensure unit classifications were robust and repeatable across the dataset, multiple acoustic parameters were measured for all units from one high-quality example of each phrase type that was present for each location/year combination. Following Dunlop *et al*. [58], Raven Pro (spectrogram parameters supplied above) was used to calculate duration, bandwidth, peak frequency, high frequency and low frequency of the fundamental frequency of each unit. The start frequency, end frequency, frequency trend (start frequency ÷ end frequency), frequency range (high frequency ÷ low frequency), number of inflections, pulse repetition rate (per second) and a qualitative name per unit were also recorded manually, following Dunlop *et al*. [58] and Garland *et al*. [42]. A random forest analysis (R package 'randomForest' [59]) was conducted in R [60] to test the agreement in qualitative unit classification using qualitative name as the dependent variable (mtry = 3, 1000 trees grown) [42,61]. The random forest classified each unit based on its quantitative parameter values and resulted in a confusion matrix revealing which units were classified together, what variable was most informative, which unit types were often mis-classified and how this corresponded to qualitative classification (electronic supplementary material, S2, table S2.2). The out-of-bag (OOB) error rate was 15.84% indicating a high level of agreement between qualitative and quantitative unit classification, confirming that unit classifications were robust and repeatable.

## 2.5. Assigning unit sequences to phrase types using the Levenshtein distance

Phrase and theme assignments were undertaken following the methods of Garland *et al*. [42], which have been used in previous studies [37,42,62,63]. During transcription, unit sequences were qualitatively assigned to phrases. Phrases were assigned a number (e.g. phrase 1, phrase 2), with small variations within a phrase (e.g. phrase types) denoted as 'A', 'B' and so forth, if they occurred consistently (electronic supplementary material, table S3.1). To check the robustness of phrase type assignment across all songs, regardless of location or year they were recorded, a normalized version of the

**Table 2.** Transcribed song data from 21 recording events from 2015 to 2017 from two breeding populations: New Caledonia and East Australia. The recording ID (X, Y or Z) identifies each analysed song per location/year combination. Duration = duration of transcribed song.

| recording location | year | recording ID | date | duration (min) | migration stage |
|---|---|---|---|---|---|
| New Caledonia | 2015, song type A | X | 17 July | 48 | breeding ground |
| | | Y | 18 July | 32 | |
| | | Z | 23 Aug | 47 | |
| New Caledonia | 2015, song type B | X | 5 Aug | 45 | breeding ground |
| | | Y | 6 Aug | 51 | |
| | | Z | 18 Aug | 52 | |
| New Caledonia | 2016 | X | 25 July (a.m.) | 60 | breeding ground |
| | | Y | 25 July (p.m.) | 55 | |
| | | Z | 28 July | 55 | |
| New Caledonia | 2017 | X | 5 Aug | 53 | breeding ground |
| | | Y | 24 Aug | 45 | |
| | | Z | 29 Aug | 52 | |
| East Australia | 2015 | X | 18 July | 30 | end of northbound migration |
| | | Y | 23 July | 40 | |
| | | Z | 17 Sep | 44 | start of southbound migration |
| East Australia | 2016 | X | 25 Sep | 40 | start of southbound migration |
| | | Y | 27 Sep | 50 | |
| | | Z | 19 Oct | 51 | |
| East Australia | 2017 | X | 6 July | 41 | end of northbound migration |
| | | Y | 17 July | 44 | |
| | | Z | 18 July | 44 | |

Levenshtein distance (LD) was conducted; the Levenshtein distance similarity index (LSI). The LD calculates the number of insertions, substitutions and deletions required to change one string into another string, while the LSI standardizes this comparison by dividing the LD result by the length of the longest sequence in the pair to reveal which strings are most similar (as a proportion). The LSI was calculated in R using custom-written code (package 'leven', available at http://github.com/ellengarland/leven) for every possible pair of phrase strings (i.e. a sequence of units), resulting in a theme similarity matrix (that documents within-group and between-group similarities) to verify qualitative phrase type assignments (electronic supplementary material, S3). Using the LSI to check phrase type assignment ensured a robust and repeatable classification method at this level within the song hierarchy.

Hierarchical clustering of the LSI similarity matrix of phrase strings ($n = 3183$; electronic supplementary material, S3, Dataset_S1) was conducted to check phrase type assignments, and visually displayed as a dendrogram to show phrase connections. The cophenetic correlation coefficient (CCC) was calculated to compare dendrograms to determine the most appropriate clustering method for the data (considered 'good' if value greater than 0.8 [64]). A comparison of CCCs confirmed that 'average linkage' clustering generated a better representation of the connections within the data than 'single linkage' clustering (CCC = 0.92 versus CCC = 0.61, respectively). A weighted analysis was also run following Garland *et al*. [42] and produced similar results in phrase type assignment; therefore, the unweighted analysis was chosen as the CCC value was slightly higher (CCC = 0.92 compared to CCC=0.90 for the weighted analysis). Once the clustering method and weighting had been determined, hierarchical clustering of the LSI theme similarity matrix was bootstrapped (1000 times),

using the 'hclust, 'pvclust' [65] and 'leven' packages in R to ensure the resulting dendrogram structure was stable and likely to occur. The bootstrap analysis generated approximately unbiased (AU) values; in order to support the structuring, AU values exceeding 95% were desirable.

To represent each phrase type with a single string, thereby condensing variability within phrases, the most representative unit string (median string) was obtained for each phrase type per location and year (electronic supplementary material, table S3.1); all pairwise LSI values were summed and the string with the largest value was deemed the median string as it was most similar to all other strings within the set (following Garland *et al.* [62] and [42]). Variability within each overall set is reported (i.e. within-set similarity; electronic supplementary material, table S3.1). The LSI phrase median string similarity matrix was hierarchically clustered and bootstrapped, as per the theme similarity matrix (electronic supplementary material, figure S3.1). Bootstrap analyses of both the theme similarity matrix and median string similarity matrix revealed that unit strings within phrase types were highly similar, with lower similarity between phrase types. Phrase type assignments were, therefore, considered robust.

## 2.6. Matching song sequences across locations and years

Once phrase assignments, and thus theme assignments, were confirmed, two analyses were conducted to match songs among locations, years and recordings: LSI using the ordering of themes; and Dice's similarity index (DSI) based on the presence and sharing of themes regardless of their sequence information (further details for the two methods are provided in Garland *et al.* [42]). For the LSI analysis, the order of themes making up each song were compiled into strings, with each theme identified by the identification number of its constituent phrase type (electronic supplementary material, Dataset_S2). Each theme was entered into a song string once, in sequence, regardless of whether the constituent phrase was sung once or repeated multiple times. The LSI was run on (i) the full dataset comprising all song strings to incorporate the variability (theme similarity matrix), and (ii) the median song string per recording to create a point estimate (median string similarity matrix) (electronic supplementary material, S4). Both results were average-linkage hierarchically clustered and bootstrapped (1000 times) to identify the similarities and connections among songs from each location/year/recording combination.

In addition to the LSI, DSI was also used to examine relationships among songs. DSI takes into account the number of shared phrases out of the total number of phrases present between all pairs of location/year/recording combinations in order to consider their similarity—see Garland *et al.* [63] and [42] for further details. Unlike the LSI, DSI does not consider the sequence of themes, merely their presence or absence in a song (electronic supplementary material, Dataset_S3). The DSI analysis was run in R using custom-written code (available at https://github.com/ellengarland/dice_si). As per the LSI analysis, the DSI analysis resulted in an overall similarity matrix, which was average-linkage hierarchically clustered and bootstrapped (1000 times).

# 3. Results

## 3.1. Spatio-temporal presence of humpback whale song in New Zealand

The PAMGuard detector outputs (electronic supplementary material, S1, figure S1.1) directed analyses to June, July and August 2016. Files containing humpback song were recorded between 8 June and 9 August 2016 (figure 2). It was not possible to check whether song was absent for at least five full days prior to 8 June, as acoustic recordings commenced on 4 June 2016. Humpback song was most abundant in data recorded at STB, while the least amount of song was recorded at Wairarapa, none of which was of high quality (i.e. category 2) (figure 2). On 21 July, 29 July and 1 August 2016, all files recorded at STB contained song (figure 2). Song of high enough quality to transcribe (i.e. category 2) was recorded on six occasions; three times each at STB and Cook Strait (table 1). Category 2 song was recorded at STB on 15 July 2016 (figure 2, turquoise star), but transcription of units was not possible due to concurrent singing by multiple animals.

During September and October 2016, when southbound migration might have been expected [23,30], PAMGuard detections did not include true song detections. During November and December 2016, seismic survey sound could have masked song with low SNR and, therefore, certainty in the absence of southbound humpback song is constrained (electronic supplementary material, S1).

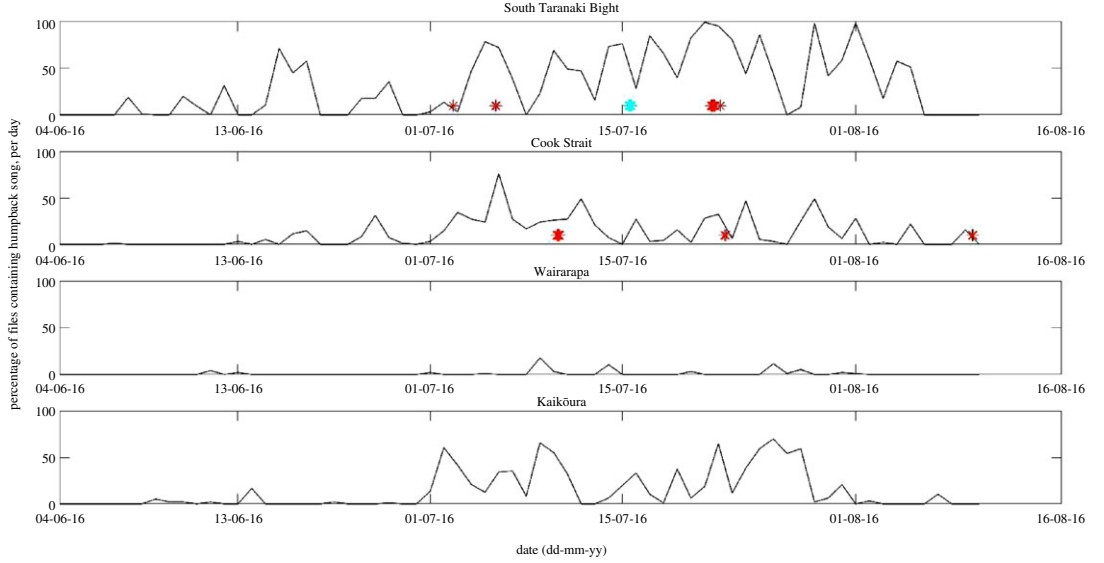

**Figure 2.** Percentage of 16 kHz files (630 s of a 900 s duty cycle) containing song (category 1 or 2) per day in 2016, per location in New Zealand. Days with high-quality song recordings (category 2) used in song matching are marked with red stars ($n = 6$). The category 2 song recorded at STB on 15 July 2016, marked with a turquoise star, was not transcribed due to concurrent singing by multiple animals.

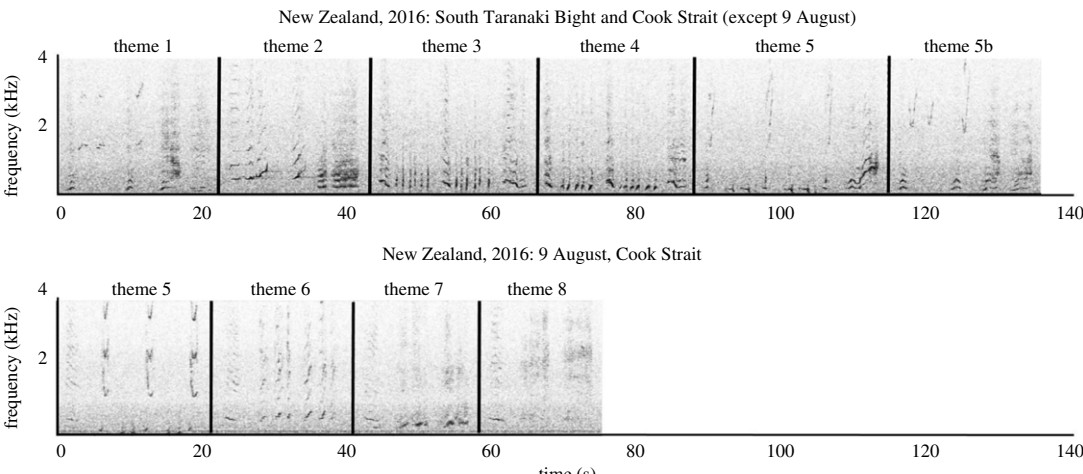

**Figure 3.** Spectrograms of the two song types recorded in central New Zealand in 2016 (FFT length 4096, Hann window, 75% overlap, displaying 4 kHz and 140 s, generated in Raven Pro 1.5). Corresponding audio files are provided for each song type (electronic supplementary material, Audio_S1 and Audio_S2).

## 3.2. New Zealand, New Caledonia and East Australia song

In total, 16 themes were present in the analysed songs, of which five themes contained multiple phrase types (electronic supplementary material, table S3.1). In the New Zealand data, five themes (1–5) were included in all transcribed songs from STB (on 2, 5 and 21 July 2016). These five themes were also present in song recorded in Cook Strait on 10 and 22 July 2016 (figure 3; electronic supplementary material, figures S4 and S5). At Cook Strait on 9 August 2016, however, only theme 5 was present, but it was accompanied by a further three themes (6, 7 and 8) that were not present in the other five New Zealand song recordings (figure 3; electronic supplementary material, figures S4 and S5). The song recorded at STB on 15 July 2016 (figure 2, turquoise star) was not suitable for transcription due to multiple whales singing concurrently with similar SNR, but was qualitatively deemed to contain themes 1–5, and not themes 6, 7 or 8.

In 2015, two song types from two different song lineages were present in New Caledonia, with no shared themes (table 3; electronic supplementary material, S4 and S5; also identified by Owen *et al.* [37]).

**Table 3.** Song themes present at each location and year. Dashes indicate year/location combinations without data. Song lineages are marked either blue or green, and evolution within a song lineage is indicated by the shade of the colour.

| year | East Australia | | New Zealand | New Caledonia |
|------|----------------|---|-------------|---------------|
| 2015 | north: 5, 6, 7, 8, 9 | | - | A: 12, 12b, 13, 13b, 14, 14b, 15 |
|      | south: 1, 2, 3, 4, 5 | | | B: 5, 5c, 6, 7, 8, 16 |
| 2016 | north:- | | 1, 2, 3, 4, 5, 5b, 6, 7, 8 | 1, 2, 3, 4, 5, 5b, 6, 7, 8 |
|      | south: 3, 3b, 9, 10, 11 | | | |
| 2017 | north: 3, 3b, 6, 9, 10, 11 | | - | 3, 3b, 4, 9, 10, 11 |
|      | south:- | | | |

The East Australian 2015 song differed between the northbound and southbound migration (table 3; electronic supplementary material, S4 and S5). Three themes (5, 6 and 8) were present in the median song string recorded during the 2015 northbound migration in East Australia; these themes were also evident in one of the 2015 New Caledonia song types (B), in the New Zealand Cook Strait song on 9 August 2016, and in song recorded in New Caledonia in 2016 (table 3; electronic supplementary material, S4 and S5). The song recorded during the 2015 southbound migration in East Australia contained the themes 1, 2, 3, 4 and 5 (table 3; electronic supplementary material, S4 and S5); only theme 5 had been present in the northbound song. These five themes were present in the majority of songs recorded in New Zealand in 2016 and were also present in 2016 New Caledonia song (table 3; electronic supplementary material, S4 and S5). Of the eight themes recorded in New Zealand in 2016, one was present in East Australia during the southbound migration in 2016 (theme 3), accompanied by three 'new' themes (9, 10 and 11) (table 3; electronic supplementary material, S4 and S5). In 2017, the southbound themes from East Australia in 2016 were repeated during the northbound migration in East Australia and were also present in New Caledonia (table 3; electronic supplementary material, S4 and S5).

## 3.3. Song similarity across the western South Pacific

### 3.3.1. Levenshtein similarity index

Hierarchical clustering and bootstrapping of all song strings revealed strong within-year and within-location similarities in song (figure 4a). Overall, clustering of song was similar when all variability among strings was considered (theme similarity matrix; electronic supplementary material, figure S6.1) as when only median strings were compared (figure 4a). The 'A' song type from New Caledonia 2015 clustered separately to all other songs (figure 4a). The southbound song from East Australia 2015 (Z) clustered with the New Zealand 2016 song from the STB and Cook Strait (excepting the song from 9 August (Z)) (figure 4a). The 9 August 2016 (Z) song recorded in Cook Strait, New Zealand, matched most strongly with the northbound song from East Australia 2015 (X and Y), and these three song types were contained in a wider cluster that included the New Caledonia 2016 song (figure 4a). At a broader level, this cluster also featured the New Caledonia 'B' song type from 2015 (figure 4a). East Australia southbound 2016, East Australia northbound 2017 and New Caledonia 2017 clustered together (figure 4a), and these were all very similar songs sharing five themes (table 3; electronic supplementary material, S4).

### 3.3.2. Dice's similarity index

When the DSI matrix was hierarchically clustered and bootstrapped, results were very similar to the LSI analysis (figure 4b). Again, the 'A' song from New Caledonia 2015 clustered separately. The New Zealand 2016 song (except Cook Strait on 9 August (CS_Z)) clustered most strongly with itself, the southbound East Australian 2015 song (Z) and the 2016 New Caledonia song. This cluster was part of a wider group that included the Cook Strait 9 August song (CS_Z), the northbound East Australia 2015 song (X and Y) and the New Caledonia 'B' song from 2015. The 2017 song from New Caledonia and northbound East Australia clustered together with the 2016 southbound song from East Australia.

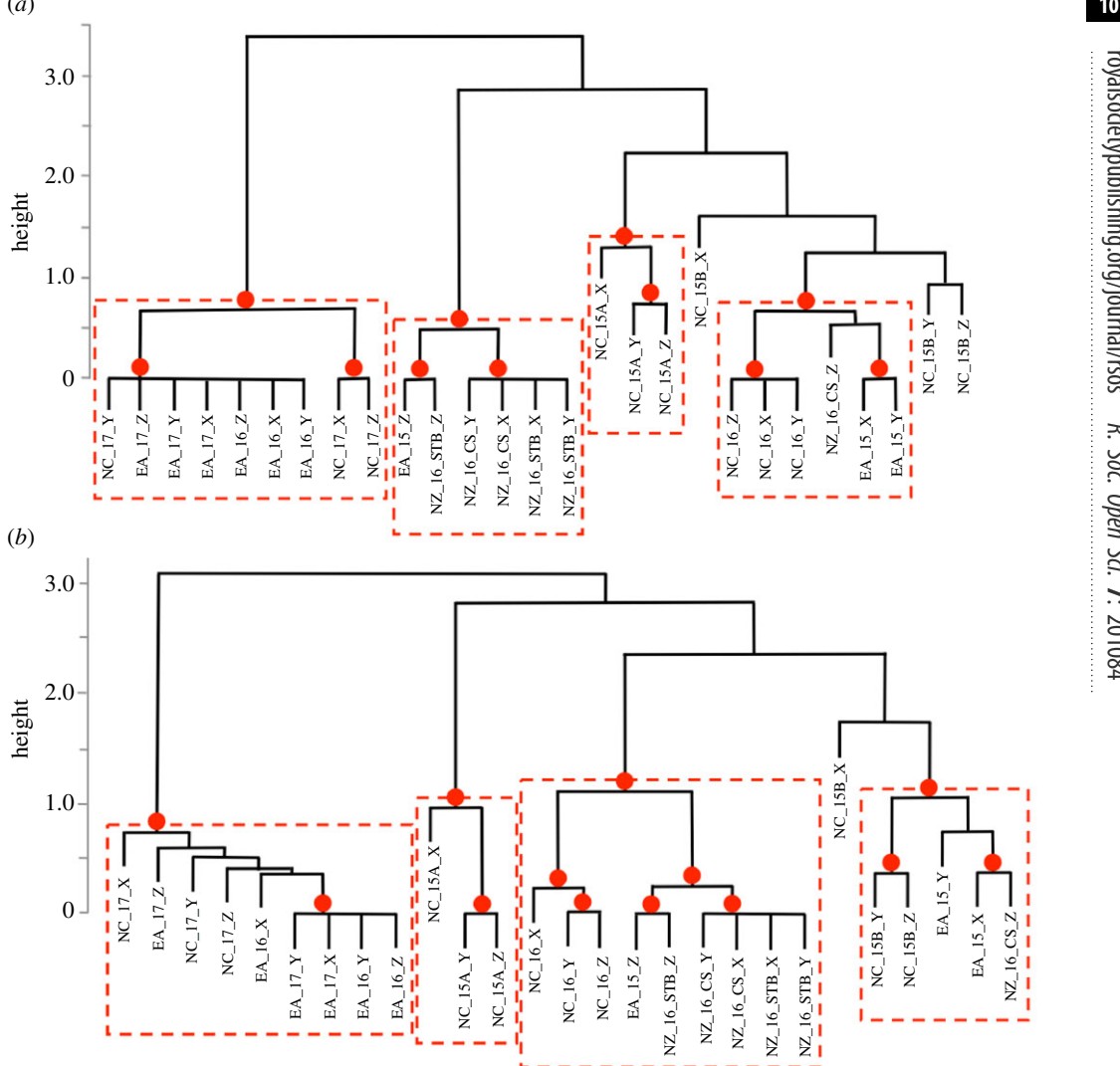

**Figure 4.** (*a*) Bootstrapped (*n* = 1000) dendrogram of average-linkage clustering of median song strings recorded at different locations and years, based on LSI analysis. (*b*) Bootstrapped (*n* = 1000) dendrogram of average-linkage clustering of theme presence and sharing among different locations and years, based on DSI. Red dots indicate AU values greater than 95% where divisions were stable and likely to occur. Red boxes indicate the resulting clusters. The labels are structured as follows: Location_Year_Sub-Location_SongIdentifier. Sub-locations are included for New Zealand 2016: STB = South Taranaki Bight; CS = Cook Strait. Two song types (A and B) were present in New Caledonia in 2015.

## 4. Discussion

Humpback whales on their northbound migration to western South Pacific breeding grounds passed through central New Zealand during June, July and early August 2016, as demonstrated by song presence. The recorded whales migrated along the east coast of the South Island of New Zealand, and continued north through Cook Strait and into the STB, or along the east coast of the North Island, although the latter route was supported by fewer song recordings. Similarities in song themes suggested that these whales most likely continued onto the breeding ground of New Caledonia, although connectivity with the breeding ground of East Australia could not be ruled out due to a data gap for northbound East Australia 2016 song. Song similarities between 2015 and 2016, and 2016 and 2017, indicated that song was most likely transmitted from East Australia to New Caledonia along shared migration routes and/or on shared feeding grounds during the austral summer. Given the occurrence of two song types in the recordings from central New Zealand, there is strong potential for song transmission to occur within this migratory corridor.

## 4.1. Spatio-temporal presence of song reveals migratory routes through central New Zealand waters

Northbound humpback whales have been observed in New Zealand waters between May and August, the austral winter [23,30]. Historically, observations of northbound whales in central New Zealand peaked from 19 June to 9 July, based on data from 1912 to 1955 [23]. This information was used to inform the timing of land-based visual surveys of migrating whales in Cook Strait over a 12 year period between mid-June and mid-July 2004–2015 [31]. Here, the data demonstrated that singing humpback whales were present in central New Zealand waters between 8 June and 9 August 2016. The timing of humpback migration through New Zealand waters varies between years [23], and no acoustic data were available prior to June 2016, but passive acoustic monitoring has enhanced our understanding of the spatio-temporal extent of the northbound migration period of whales through these waters as they recover post-whaling.

The location of the acoustic recorders in this study covered known historic and modern humpback whale migration paths through central New Zealand. The data suggest that the singing whales that passed the Kaikōura recorder on the east coast of the South Island during their northbound migration primarily travelled through Cook Strait and into the STB, rather than continuing along the east coast of the North Island, past the Wairarapa recording location. This conclusion is based on the low numbers of song detections made at Wairarapa. However, we cannot reject the hypothesis that whales detected at Kaikōura continued along the east coast of the North Island; these animals may have ceased singing, or travelled further offshore to an area that was outside of the detection range of the Wairarapa recorder. The song recorded in Cook Strait on 9 August 2016 was different to the song recorded at the same location earlier in the migratory period, demonstrating consistent use of Cook Strait by mature male whales, despite differences in song content. Song produced by humpback whales passing offshore, or to the west of New Zealand, outside of the STB, would not have been recorded in this study, and presents an area for future research.

Humpback song can function as a proxy for the presence of mature male whales, but Dawbin [66] reported temporal segregation among whales of different age and sex classes passing through New Zealand waters. As such, song may not be a strong predictor for the migratory presence of whales that do not sing (females, juveniles and silent males). Moreover, Valsecchi et al. [26] hypothesized possible sexual segregation in migratory routes of humpback whales in the South Pacific, and female or juvenile whales may traverse alternative paths. In order to investigate the composition of whales in the migratory corridor, it would be necessary to conduct passive acoustic monitoring in combination with individual-specific methodologies, such as photographic mark–recapture or genetic sampling, over the whole migratory period to understand the relationship between singing-male presence with other demographic groups [24].

There were no detections of song from humpback whales during the expected period of southbound migration (September–December), although acoustic contributions from seismic surveys meant that it was not possible to determine the absence of humpback whale song with certainty. However, humpback whale presence would not be expected in high quantities in the vicinity of the acoustic recorders during the southbound migration, as satellite tag studies have shown little evidence of southward travel near central New Zealand [18,22,32]. Whales may prefer to travel south to Antarctica along more direct routes from their breeding grounds to minimize energy costs to lactating mothers and young calves [22].

## 4.2. Song connections between New Zealand and breeding populations

Song recorded in central New Zealand during the northbound migration in 2016 matched song recorded on the New Caledonia breeding ground in 2016, and was dissimilar to song recorded in East Australia in 2016. The data, therefore, suggest that the singing humpback whales that passed through central New Zealand were on a migratory path towards New Caledonia. New Caledonian song is commonly similar to song heard at the breeding ground in Tonga within a given year [39,62], and similar song is also likely to occur at pelagic seamounts adjacent to New Caledonia where humpback whales are found during the breeding season [6,36]. Thus, the New Caledonian song considered here may act as a proxy for western Oceania breeding grounds. Nonetheless, song similarity presented here is likely to be indicative of a robust connection between the migratory corridor (New Zealand) and the breeding ground (New Caledonia). Indeed, humpback whales sampled in New Zealand are most genetically

similar to the breeding population in New Caledonia [4,24] and photographic mark–recapture studies have shown strong links [33].

That said, sample sizes in this study are small and no song data were available from the northbound migration in East Australia in 2016 to compare with song recorded in New Zealand in 2016. In 2015, song evolved in East Australia during the breeding season, and the same could have occurred during 2016, meaning that the 2016 East Australia song presented here, from the southbound migration, would have been dissimilar to the northbound 2016 East Australia song and potentially the same as the song recorded in central New Zealand. The New Zealand migratory corridor has been connected to the East Australian breeding ground in recent times via song matching [40] and other methods [8,24,31,33,34], and it is possible that some whales that migrated through central New Zealand could have travelled on to the breeding ground of East Australia. Connectivity between the New Zealand migratory corridor and the breeding ground of East Australia cannot be conclusively investigated within the present study.

The results presented here support the hypothesis that the singing humpback whales recorded during the northbound migration through central New Zealand travelled on to the breeding grounds of western Oceania, but in acknowledging the caveats of the data outlined above, there is also a possibility that some whales travelled to the breeding ground of East Australia. Whales that traversed other migration routes past New Zealand, outside of the recording area, or those that travelled through the recording area without detection, could have been travelling to different breeding areas.

## 4.3. Song evolution, revolution and transmission

In order for song to be transmitted between whales, individuals must be within the limits of acoustic contact, which has been estimated to be approximately 20 km [45]. This could occur when animals move between breeding grounds (within or between years), while on shared feeding grounds, or while following shared or partially shared migration routes [45].

In 2015, the 'B' song from New Caledonia was the same song produced by northbound East Australian whales. The 'A' song that was initially present in New Caledonia in 2015 was mostly replaced by the 'B' song (see Owen *et al.* [37] where these two songs are referred to as '1b' and '2', respectively). As there were no overlapping themes between the 'A' and 'B' song types (and indeed song lineages), the transition was an example of a cultural revolution [50] within the New Caledonia breeding area in 2015. It is possible that the 'B' song was introduced to New Caledonia by transient East Australian males that crossed the Coral Sea to New Caledonia during the breeding season, but to date, there is only evidence of the reverse movement [6,67]. Alternatively, a secondary group of whales may have arrived at the New Caledonia breeding ground, from another migratory corridor, having received greater exposure to the East Australian 2015 song. Also during 2015, the song produced during northbound migration in East Australia differed from the song recorded at the end of the breeding season. As the two song types shared a common theme, they are indicative of rapid and extensive song evolution within a single song lineage [46,68] that occurred in East Australia during the breeding season. Small sample sizes constrain the assumption that all singing whales in East Australia produced the evolved song by the end of the 2015 breeding season.

During northbound migration in 2016, the song recorded in New Zealand, and subsequently in New Caledonia, contained themes which had been present during the southbound migration past East Australia in 2015. Some of these themes had been present in New Caledonia in 2015 as the 'B' song type, but it is probable that the additional East Australian themes from 2015 were transmitted to New Caledonia whales during a period of acoustic contact (*sensu* [45]). The Balleny Islands, directly south of New Zealand, are a known feeding ground for East Australian whales [17,32]. Whales from New Caledonia also frequent the Balleny Islands, within acoustic contact of animals from the East Australian breeding ground, and this is a possible location for song transmission between the two populations while on shared feeding grounds [17,19].

A shared, or partially shared, southbound migratory route exists along the east coast of Australia as recently highlighted by a few New Caledonian whales that crossed the Coral Sea and migrated south with the East Australian migratory cohort [6,67]. This may provide opportunity for New Caledonian whales to learn the East Australian song which could then be transmitted within the New Caledonian population the following breeding season. The present study also reveals the potential for song sharing in the New Zealand migratory corridor during northbound migration. While the data did not conclusively demonstrate the presence of both East Australian and New Caledonian whales in central New Zealand, there is genetic and photographic evidence that whales from both

breeding populations are the primary users of this migratory corridor [4,11,24,31,33,34], and the presence of two song types in the central New Zealand recordings indicated the potential for vocal learning to occur in this migratory corridor. Hybrid songs [69] provide direct evidence of song sharing, but these occur rarely, and recording them in the small sample sizes presented here would have been highly unlikely. Hybrid songs have previously been noted during southbound migration past the Kermadec Islands, New Zealand [37], providing further evidence for song sharing among humpback whales around New Zealand.

The possible transmission points discussed above may also underpin the transmission of East Australian 2016 song to whales in New Caledonia for the 2017 breeding season. The data presented here provide strong support for song transmission during northbound migration. While this study does not provide explicit evidence for song transmission within the New Zealand migratory corridor, the presence of two song types illustrated a potential location for song sharing to occur. The recorded songs provided further evidence of an eastward transmission of song across the South Pacific over consecutive years, as has been widely reported [19,39,62]. Data from additional years, as well as a future focus on song recording along migratory corridors, would help to further understanding of song transmission.

# 5. Conclusion

Autonomous passive acoustic instruments deployed in the marine environment of central New Zealand recorded singing humpback whales on their northward migration to western South Pacific breeding grounds from June until early August. Song detections revealed that the predominant migration route from the east coast of the South Island was through Cook Strait, rather than along the east coast of the North Island. The recorded song implied the most likely breeding ground destination of these whales was New Caledonia, although connectivity to East Australia could not be ruled out. Song themes from whales on their southbound migration past East Australia in 2015 were heard in New Caledonia in 2016, implying that song transmission occurred on summer feeding grounds or on a shared migratory route between the breeding seasons of 2015 and 2016. The presence of two song types in New Zealand recordings in 2016 demonstrated a potential for song sharing to occur in this migratory corridor. The results presented here greatly enhance our knowledge of humpback whale movements through central New Zealand, and provide information about song transmission and migration routes of western South Pacific humpback whales. While many questions remain unanswered, this study demonstrates the applicability of passive acoustic monitoring to provide a non-invasive, cost-effective methodology to study western South Pacific humpback whales, of which, some populations are still recovering from the effects of commercial whaling.

Ethics. All experiments are in compliance with the guidelines for the treatment of animals in behavioural research and teaching, and within the permitted conditions required under law.

Data accessibility. The datasets supporting this article have been uploaded as part of the electronic supplementary material: Dataset_S1_phrases.txt, Dataset_S2_Leven.txt, Dataset_S3_Dice.txt.

Authors' contributions. V.E.W. conducted all analyses and wrote the manuscript, under the supervision of E.C.G. and R.C. M.N. and C.G. contributed acoustic data. All authors interpreted data and critically revised the manuscript. All authors approve the final manuscript.

Competing interests. We have no competing interests.

Funding. V.E.W. is funded by a University of Auckland Doctoral Scholarship and the Woodside Marine Mammal Research Grant awarded by Woodside Energy. E.C.G. is funded by a Royal Society University Research Fellowship. Data collection in New Zealand was conducted by the National Institute of Water and Atmospheric Research (NIWA), Wellington, supported by funding from OMV New Zealand Ltd, Chevron New Zealand Holdings LLC and Marlborough District Council. New Caledonian recordings were funded by a grant to Jennifer Allen from the Sea World Research and Rescue Foundation Inc. East Australian recordings were funded by the Joint Industry Programme, E&P Sound and Marine Life as part of the BRAHSS project (2015) and by the Cetacean Ecology and Acoustics Lab (2016, 2017).

Acknowledgements. Thanks to the New Zealand acoustic deployment and retrieval teams from the National Institute of Water and Atmospheric Research, Wellington: Sarah Searson, Mike Brewer, Olivia Price, Fiona Elliott and the crews of the RVs Tangaroa, Kaharoa and Ikatere. Thanks to Jennifer Allen (Cetacean Ecology and Acoustics Lab, University of Queensland) for collecting the New Caledonian recordings, and the Department of Environment of the Southern Province of New Caledonia for their help in deploying and recovering the acoustic equipment. Further thanks to Craig Radford (University of Auckland) and Kimberly Goetz (NOAA) for feedback that improved the manuscript, and again to Jennifer Allen for insights into song coding.

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
