## [Reviewer comments · Royal Society Open Science]

Review History

RSOS-201084.R0 (Original submission)

Review form: Reviewer 1 (Marc Lammers)

Is the manuscript scientifically sound in its present form?

Yes

Are the interpretations and conclusions justified by the results?

Yes

Is the language acceptable?

Yes

Do you have any ethical concerns with this paper?

No

Have you any concerns about statistical analyses in this paper?

Yes

Recommendation?

Accept with minor revision (please list in comments)

Comments to the Author(s)

Well written paper. Nice work!

Decision letter (RSOS-201084.R0)

Dear Ms Warren

On behalf of the Editors, we are pleased to inform you that your Manuscript RSOS-201084 "Migratory insights from singing humpback whales recorded around central New Zealand" has been accepted for publication in Royal Society Open Science subject to minor revision in accordance with the referees' reports. Please find the referees' comments along with any feedback from the Editors below my signature.

Please submit your revised manuscript and required files (see below) no later than 7 days from today's (ie 20-Oct-2020) date. Note: the ScholarOne system will 'lock' if submission of the revision is attempted 7 or more days after the deadline. If you do not think you will be able to meet this deadline please contact the editorial office immediately.

on behalf of Dr Asha de Vos (Associate Editor) and Kevin Padian (Subject Editor)
openscience@royalsociety.org

Editors' Comments to Author:

We had difficulty finding reviewers for the manuscript, unfortunately, which explains the length of time in process so far. Fortunately we have one very positive review that raises some questions that we ask you to respond to with your revised manuscript. Best wishes.

Reviewer comments to Author:

Reviewer: 1

Comments to the Author(s)

Well written paper. Nice work!

(See attached file)

===PREPARING YOUR MANUSCRIPT===

- one version identifying all the changes that have been made (for instance, in coloured highlight, in bold text, or tracked changes);
- a 'clean' version of the new manuscript that incorporates the changes made, but does not highlight them.

This version will be used for typesetting.

===PREPARING YOUR REVISION IN SCHOLARONE===

Author's Response to Decision Letter for (RSOS-201084.R0)

See Appendix A.

Decision letter (RSOS-201084.R1)

Dear Miss Warren,

It is a pleasure to accept your manuscript entitled "Migratory insights from singing humpback whales recorded around central New Zealand" in its current form for publication in Royal Society Open Science. The comments of the reviewer(s) who reviewed your manuscript are included at the foot of this letter.

on behalf of Dr Asha de Vos (Associate Editor) and Kevin Padian (Subject Editor)
openscience@royalsociety.org

Associate Editor Comments to Author (Dr Asha de Vos):

Associate Editor

Comments to the Author:

Thank you for your patience but also quick turnaround. This work is great and I look forward to see it published at long last.

Appendix A

Leigh Marine Laboratory
The University of Auckland
PO Box 349
Warkworth, New Zealand

160 Goat Island Road, Leigh
Telephone: +64 4 – 386 0526
Cellphone: +64 21- 0828 5241
Email: vwar775@aucklanduni.ac.nz

22nd October 2020

Dear Dr de Vos and Mr Padian,

Re: Manuscript RSOS-201084

My co-authors and I would like to thank you for considering our above-referenced manuscript. We are very grateful for the positive and constructive feedback from the reviewer. We have studied the review comments carefully and have revised the manuscript accordingly. A detailed summary appears below, with our responses given in bold type.

One further alteration has been made within the Supplementary material, as it was noted that the caption at the beginning of Supplementary section S4 referred to spectrograms, when in fact the spectrograms are in section S5. This has now been resolved.

Yours sincerely,

Victoria Warren

Reviewer 1

Review of Warren, Constantine, Noad, Garrigue and Garland (Royal Society) “Migratory insights from singing humpback whales recorded around central New Zealand.” The paper describes an effort to use acoustic monitoring of song to study the migration path of humpback whales as they transit past New Zealand on their way to low latitude breeding grounds. Overall, authors produce compelling evidence indicating that humpback whales transit through central New Zealand and that the region may represent an important geographical area for song exchange among breeding populations. I do not entirely agree with the author’s conclusion about the relative importance of the Cook Strait to an offshore transit route past the North Island (see below), but that is a relatively minor point of disagreement. The manuscript is well written and presented and the data are clear (for the most part). I therefore recommend the manuscript be published.

Specific Comments

Introduction: The introduction is very well written. No specific comments or suggested edits.

Materials and Methods: The section is well written. I must admit a lack of familiarity with some of the statistical approaches used (e.g. Levenshtein Distance, cophenetic correlation coefficient), so I cannot provide a critical review of their appropriateness. Someone with a deeper background will be able to provide a better assessment. However, I do note that the methods used are not novel and have been applied in previous published works on whale song analysis, so I am willing to give the authors the benefit of the doubt on them.

Results: The results are well explained and supported.

Discussion:

- P.7, 40-41: I don't think you can say where "most" whales passed. Cook Strait presents a bottleneck with whales likely passing very close to the recorder there and probably also at STB. Whales not going through Cook Strait may well have been traveling well offshore and out of detection range of the Wairarapa recorder. Thus, it is ill advised to speculate on the relative amount of whales using the two paths.

- **We have rewritten this sentence as follows: 'The recorded whales migrated along the east coast of the South Island of New Zealand, and continued north through Cook Strait and into the South Taranaki Bight (STB), or along the east coast of the North Island, although the latter route was supported by fewer song recordings.'**

- P.9, 18-20: The data support the conclusion that whales don't transit near the coastline in large numbers. However, given the bottleneck quality of Cook Strait compared to the open ocean expanse east of the Wairarapa coast, I think even modern and whaling era comparisons are likely biased. Why would whalers go search for a needle in a haystack in offshore waters, when you have whales threading the needle at Cook Strait? I think a migration route further offshore along the North Island's east coast cannot be excluded by your results.

- **We have removed the sentence pertaining to whaling effort in the two regions.**
- P. 9, 36-38: It's a bit unsatisfying to simply abandon the point there. Why only a northbound transit through central New Zealand? Any speculation?
 - **We have added the following sentence to the end of the paragraph: 'Whales may prefer to travel south to Antarctica along more direct routes from their breeding grounds to minimise energy costs to lactating mothers and young calves (22).'**

Again, we would like to extend our sincere thanks to the reviewer for these constructive suggestions which have improved the manuscript.